# Immune Cells in Hyperprogressive Disease under Immune Checkpoint-Based Immunotherapy

**DOI:** 10.3390/cells11111758

**Published:** 2022-05-27

**Authors:** Zhanqi Wei, Yuewei Zhang

**Affiliations:** 1School of Medicine, Tsinghua University, Haidian District, Beijing 100084, China; wei-zq15@mails.tsinghua.edu.cn; 2Hepatopancreatbiliary Center, Tsinghua University Affiliated Beijing Tsinghua Changgung Hospital, Changping District, Beijing 102218, China

**Keywords:** immunotherapy, hyperprogressive disease, immune cells, immune checkpoint inhibitors

## Abstract

Immunotherapy, an antitumor therapy designed to activate antitumor immune responses to eliminate tumor cells, has been deeply studied and widely applied in recent years. Immune checkpoint inhibitors (ICIs) are capable of preventing the immune responses from being turned off before tumor cells are eliminated. ICIs have been demonstrated to be one of the most effective and promising tumor treatments and significantly improve the survival of patients with multiple tumor types. However, low effective rates and frequent atypical responses observed in clinical practice limit their clinical applications. Hyperprogressive disease (HPD) is an unexpected phenomenon observed in immune checkpoint-based immunotherapy and is a challenge facing clinicians and patients alike. Patients who experience HPD not only cannot benefit from immunotherapy, but also experience rapid tumor progression. However, the mechanisms of HPD remain unclear and controversial. This review summarized current findings from cell experiments, animal studies, retrospective studies, and case reports, focusing on the relationships between various immune cells and HPD and providing important insights for understanding the pathogenesis of HPD.

## 1. Introduction

Immunotherapy has emerged as a new and developing method of tumor treatment in recent years and is now widely used in clinical practice [1]. A total of 10 immune checkpoint inhibitors (ICIs) targeting programmed cell death protein 1 (PD-1) or its ligand programmed cell death ligand 1 (PD-L1), such as pembrolizumab, nivolumab, and atezolizumab, as well as one ICI targeting cytotoxic T-lymphocyte-associated antigen 4 (CTLA-4), ipilimumab, have been approved for clinical application in the United States and China [2]. However, many studies have reported that the reactivities of patients treated with ICIs differ significantly. Firstly, a long duration of tumor shrinkage occurs in 10% to 30% of patients treated with ICIs, which is much higher than that achieved with other tumor treatments, and overall survival (OS) is improved unprecedentedly [3]. Secondly, under ICI treatment, some patients exhibit mixed responses or pseudoprogression, which is defined as an initial surge in the tumor burden, followed by tumor shrinkage [4]. Finally, 4% to 29% of patients cannot benefit from ICIs. Instead, they experience faster and more aggressive tumor progression than expected, which is known as hyperprogressive disease (HPD) [5]. As shown in Table 1, the correlations between HPD and ICIs have been shown across multiple types of tumor. A number of studies have proved the influence of HPD on OS, progression-free survival (PFS), and the rate of tumor progression [6,7,8,9,10,11,12,13,14,15,16,17,18,19,20,21,22,23,24,25]. As ICIs become more prevalent, improved knowledge of HPD is urgently needed to precisely determine which patients should receive immunotherapy. MDM2 and EGFR mutations have only been considered as potential risk factors, rather than causal factors, because these gene mutations predispose tumors to progression. Thus, current studies are committed to determining the exact cause of HPD [26]. In addition, ICI-based immunotherapy is designed to restore immune cell attack against tumors by blocking PD-1 or CTLA-4. Thus, immune cells certainly play a crucial role in the development of HPD. As more data become available, the reasons behind this phenomenon may be uncovered, allowing the application of preemptive measures to mitigate HPD. The purpose of this review is to provide an update about HPD and explain the possible mechanisms by which ICIs induce HPD by influencing various types of immune cells.

## 2. CD8^+^ T Cells

The immune checkpoint blockade (ICB) was originally designed to restore T-cell attack against tumor cells by blocking the primary negative regulator of T-cell function, PD-1, with ICIs [27]. Therefore, the responses of tumor patients to ICIs are influenced by the infiltration of activated CD8^+^ T cells into the tumors. Tumors filled with T cells are generally considered to be more sensitive to ICIs, while tumors lacking tumor-infiltrating lymphocytes (TILs) have a low absent response to ICIs [1]. Because CD8^+^ T cells play a crucial role in immunotherapy, we speculate that CD8^+^ T cells may also contribute to the development of HPD.

### 2.1. Exhausted CD8^+^ T (Tex) Cells

In patients with chronic infection and tumors, long-term exposure to persistent antigens and inflammation causes continuous CD8^+^ T cell stimulation to the point that the cells gradually lose effector function and memory T-cell characteristics, which is a condition known as T-cell exhaustion [28]. Functionally different from effector T cells and memory T cells, Tex cells are characterized by effector function loss, continuously increased regulation of inhibitory receptor (IR) expression, changes in epigenetic and transcriptional profiles, and changes in metabolic patterns [29]. T-cell exhaustion is one of the main factors of immune dysfunction in cancer patients [30]. In multiple tumors, Tex cells exhibit dysfunction in producing effector cytokines, limiting the immune response to tumor cells [31,32,33,34].

It has been reported that PD-1^−^CD8^+^ T cells lack terminal differentiation and proliferative ability and are driven into an exhausted state [35]. Kim et al. [11] demonstrated that a lower number of CCR7^−^CD45RA^−^ T cells, which are effector or memory cells, combined with a higher number of TIGIT^+^ T cells, which are severely depleted tumor-specific CD8^+^ T cells, was associated with the development of HPD and poor survival in tumor patients. However, Wartewig et al. [36] observed that anti-PD-1/PD-L1 treatment could induce HPD in T-cell non-Hodgkin’s lymphoma mice. Researchers demonstrated that PD-1 inhibited carcinogenic signaling by increasing the level of PTEN while decreasing PI3K/AKT and NF-κB signaling pathway activity. Hence, anti-PD-1 antibodies are able to rapidly accelerate T-cell growth. Ratner et al. [37] reported HPD after nivolumab treatment in three patients with chronic, smoldering acute lymphoma and adult T-cell leukemia.

Stein et al. [38] analyzed the correlation between TILs and tumor cells. Since T cells that infiltrated into breast tumors showed low expression of granzyme B [39] and high expression of PD-1 [40], they focused on cognate nonlytic antigen-specific interactions between CD8^+^ T cells and tumor cells. They proved that CD8^+^ T cells were activated by and interacted with breast cancer cells but could not kill them, thereby inducing the expression of pluripotency-associated genes in tumor cells and thus inducing cancer stem cell (CSC) formation [41,42]. They further demonstrated that T cells deprived of cytotoxic activity could reshape tumor cells after symbiotic interaction with CSCs. They hypothesized that ICIs could promote these nonlytic interactions and lead to hyperprogression through the dedifferentiation of tumor cells. This mechanism is the basis of the epigenetic resistance hypothesis, suggesting that CSCs contribute to HPD, but the associated biological mechanism remains to be discovered.

It is worth mentioning that Jiang et al. [43] introduced homeostasis of the activated and exhausted CD8^+^ T cells in tumor microenvironment (TME). They explained the curative effect of ICIs via the expansion and differentiation of progenitor Tex cells and believed that the depletion of the T cell progenitors and the effector function of Tex cells were the causes of the ineffectiveness of immune monotherapy under exorbitant tumor burden. Therefore, they proposed that ICIs should be combined with vaccines, chemotherapy, radiotherapy, targeted therapy, and cytokines to reduce tumor burden and expand the progenitor T cell pool.

### 2.2. Immune Checkpoint Compensation Mechanisms

PD-1 blockade triggers compensation mechanisms that suppress CD8^+^ T-cell activity [44,45,46]. This reverse regulatory feedback may counteract the benefits of anti-PD-1 treatment based on T-cell activation and in turn establish an enhanced immunosuppressive TME, laying the foundation for the progression of HPD [47]. Huang et al. [44] reported that TILs isolated from ovarian cancer-model mice simultaneously expressed multiple ICI receptors, such as PD-1, CTLA-4, and lymphocyte-activation gene 3 (LAG-3), which are involved in the creation of a highly immunosuppressive TME. Interestingly, PD-1 blockade confers compensatory enhanced expression of LAG-3 and CTLA-4 in CD8^+^ T cells, negating the benefits of ICB. Koyama et al. [48] analyzed the TME in mice with lung adenocarcinoma and in two non-small-cell lung cancer (NSCLC) patients treated with anti-PD-1 agents. They observed T-cell immunoglobulin and mucin domain-3 (TIM3) overexpression in tumor-permeable cytotoxic CD8^+^ T cells after immunotherapy failure. TIM3 expression level was significantly related to the duration of PD-1 blockade. These results suggest that anti-PD-1/PD-L1 antibodies activate compensatory immunosuppression and immune escape processes and may lead to HPD. Thus, the combined application of multiple ICIs may be more effective in inhibiting tumor progression than anti-PD-1 monotherapy. However, the mechanisms of interactions between different immune checkpoints need to be further explored.

### 2.3. IL-10/IL-10R

Sun et al. [49] reported that circulating tumor-specific CD8^+^ T cells in advanced melanoma with high PD-1 expression also have high expression levels of IL-10R and that the level of IL-10R expression can be further increased by PD-1/PD-L1 blockade, making these T cells more sensitive to IL-10. Therefore, the PD-1 blockade-induced secretion of IL-10 by innate immune cells, such as monocytes and the upregulation of IL-10R expression in T cells, may lead to severe inhibition of the antitumor activity of CD8^+^ T cells, thus forming a vicious cycle of immune escape that is conducive to tumor growth. In addition, the researchers also reported that the inhibition of IL-10 enhanced the role of anti-PD-1 antibodies in amplifying tumor-specific CD8^+^ T cells, thereby enhancing their antitumor activity. These findings further prove that IL-10 and IL-10R are involved in the development of HPD.

## 3. CD4^+^ T Cells

Immunotherapy research has often focused on CD8^+^ T cells because of their ability to eliminate tumor cells. However, CD4^+^ T cells have attracted attention in the field because they are not only crucial for promoting CD8^+^ T cell functions, preventing CD8^+^ T cell depletion or inducing CD8^+^ T cell memory, but also able to directly or indirectly kill tumor cells [50].

### 3.1. Regulatory T (Treg) Cells

In addition to effector cells, the T lymphocyte family includes an immunomodulatory subgroup called Treg cells, whose role is to negatively regulate other immune cells, prevent the overactivation of the immune response, and play a role in a wide range of diseases, such as allergies, chronic infections, and parasitic infections [51]. However, the presence of Treg cells is disadvantageous to hosts with tumors because they limit an effective antitumor immune response. Kamada et al. [52] reported that the proportions of effector regulatory T (eTreg) cells/CD8^+^ T cells, Ki67^+^ Treg cells/Ki67^+^CD8^+^ T cells, and Ki67^+^ Treg cells decreased significantly in non-HPD patients after treatment with anti-PD-1 antibodies, while these proportions in HPD patients remained stable or even increased slightly. This finding suggested that if the number of CD8^+^ T cells is insufficient to overcome Treg cells, the possibility of HPD development is greatly increased. Furthermore, Treg cells have also been shown to express immune checkpoints, such as PD-1; thus, Treg cells can also be targeted by anti-PD-1 agents [53]. Researchers have observed that knocking out PD-1 in Treg cells or blocking PD-1 with monoclonal antibodies (mAbs) caused Treg cells to gain a stronger proliferative ability and a stronger immunosuppression ability, thus leading to a stronger ability to promote tumor growth. This finding suggested that PD-1^+^ Treg cells play a key role in anti-PD-1 treatment-mediated HPD in advanced gastric cancer. In addition, Ratner et al. [37,54] demonstrated that nivolumab led to rapid progression in patients with adult T-cell leukemia/lymphoma (ATLL). They identified a novel relationship between tumor-resident Tregs and ATLL cells and revealed the tumor suppressive effect of PD-1 in ATLL.

Furthermore, in Treg cells treated with PD-1 blockade, the expression of immune checkpoints is upregulated, and the immunosuppressive function is enhanced. Thus, the antitumor immunity of some patients after anti-PD-1 treatment is not enhanced but greatly weakened, which leads to the occurrence of HPD. Interestingly, CTLA-4 was found to be strongly expressed in Treg cells [55]. Anti-CTLA-4 treatment increased the presence of Ki67^+^ Treg cells [52]. Furthermore, the combination of anti-CTLA-4 antibodies and anti-PD-1 antibodies was associated with a lower incidence of HPD than other ICI combinations, and CTLA-4, OX-40, or CCR4-targeted therapy might be a strategy for preventing HPD through Treg cell consumption [56]. In addition, selective PD-1/PD-L1 inhibition may lead to tumor immune evasion and accelerate tumor growth by increasing the number of Treg cells infiltrating and circulating in the tumor [57].

### 3.2. Other Subsets of CD4^+^ T Cells

We should consider that other CD4^+^ T cells beyond Treg cells are equally important. CD4^+^CD28^−^ T cells are a cell subpopulation with unique biological effects that frequently appear in some autoimmune diseases [58]. Due to their lack of CD28, which is necessary for a cell-specific immune response and the most important costimulatory molecule on the T-cell surface, these unique cells not only have abnormal immune function but also have the characteristics of autoreactivity, massive expansion, and a long lifespan [59]. Arasanz et al. [60] found that CD4^+^CD28^−^ T cells in the peripheral blood of lung cancer patients with HPD were amplified after PD-1 treatment, and high tumor growth dynamics scores were associated with the presence of CD4^+^CD28^−^ T-cell subsets in patients with HPD.

In addition, Zappasodi et al. [61] observed in melanoma mice that a subset of CD4^+^Foxp3^−^PD-1^high^ T cells can perform immunosuppressive functions similar to those of Treg cells, but their RNA expression levels may be more similar to those of follicular helper T (Tfh) cells. Interestingly, while anti-PD-1 treatment reduced the numbers of these cells, anti-CTLA4 treatment increased their intratumoral abundance. This result suggests that this cell subpopulation could also respond to ICB, proliferate under anti-CTLA-4 treatment, and acquire negative regulatory immune properties, which might contribute to the development of HPD (Figure 1).

### 3.3. Exhausted CD4^+^ T Cells

Another potential mechanism of HPD is the correlation between exhausted CD4^+^ T cells and anti-PD-1 treatment. The current understanding of CD4^+^ T cell exhaustion is obviously insufficient. However, the negative effects of CD4^+^ T cell exhaustion on proliferation, cytokine production, B-cell help, and CD8^+^ effector functions have been reported. Furthermore, exhausted CD4^+^ T cells upregulate immune-regulatory proteins, such as TIM3 and PD-1, paralleling phenotypes observed in exhausted CD8^+^ T cells [62]. Unlike non-HPD patients, HPD patients showed abnormal dilation of peripheral exhausted memory CD4^+^ T cells after the initial administration of anti-PD-1/PD-L1 antibodies [60]. Arasanz et al. [60] monitored peripheral blood mononuclear cells (PBMCs) in NSCLC patients treated with anti-PD-1/PD-L1 antibodies, and peripheral exhausted CD4^+^ T-cell proliferation was observed in patients with HPD. They proposed that the rapid expansion of peripheral CD28^−^CD4^+^ T cells is an early distinguishing feature of ICIs-induced HPD in NSCLC. Although the role of exhausted CD4^+^ T cells is not fully understood, these studies provide important evidence that these cells might also contribute to the progression of HPD.

### 3.4. IFN-γ

While IFN-γ is considered to be a key factor in antitumor immunity [63,64], Xiao et al. [65] demonstrated that IFN-γ could promote immune escape and papilloma development by enhancing a Th17-associated inflammatory reaction. Thus, IFN-γ can promote either antitumor immunity or immune escape according to the pathological background and the level of selective stress [66]. Sakai et al. [67] reported that in a mouse model of *Mycobacterium tuberculosis* infection, PD-1^−^ led to the extensive penetration of CD4+ T cells into the lung parenchyma and the production of large amounts of IFN-γ, causing rapid disease progression, compared with that observed in wild-type mice. In addition, mutations in genes encoding IFN-γ signaling pathway components, such as IFN-γ receptor and JAK1/2, have been identified as potential mechanisms of resistance against anti-PD-1/PD-L1 and anti-CTLA-4 antibodies [64,68]. Champiat et al. [40] noted that T-cell behavior in the TME under ICB may be affected by mutations that affect the IFN-γ signaling pathway, particularly mutations in JAK1/2. JAK1/2 mutations have been proven to be associated with primary resistance to ICIs [69]. In addition, it has been reported that IFN-γ-induced interferon regulatory factor 8 (IRF-8) binds to its promoter and induces MDM2 overexpression [70,71]. MDM2 is a protein involved in p53 degradation and inhibition, and its amplification is often observed in HPD patients [70].

## 4. Monocytes

Monocytes are necessary innate immune cells which circulate in the blood and travel to tissues at sites of inflammation or infection [72]. Different monocyte subsets have completely different functions during tumor development, contributing to the activation of protumor and antitumor immune responses [73]. In addition, monocytes are the main source of macrophages and dendritic cells (DCs), which are two important components of the TME [73].

Although the expression of PD-1 in monocytes is much lower than that in T cells, monocytes can express PD-1 [74,75]. The function of PD-1 in monocytes appears to be similar to that in other types of immune cells, with PD-1 largely suppressing immune function. In fact, PD-1 is thought to negatively affect IL-12 secretion in monocytes [76,77], and anti-PD-1 treatment reverses monocyte dysfunction [78]. PD-1 expression levels in monocytes are regulated by different inflammatory stimuli; for example, IL-10 has been described as an inducer of PD-1 expression [79,80]. However, when PD-1 in monocytes is triggered by specific antibodies, it causes the monocytes to release large amounts of IL-10 [81,82]. As a well-known anti-inflammatory cytokine, IL-10 inhibits antitumor immune responses by inhibiting a variety of effector molecules and tumor cells [83].

Lu et al. [15] examined 56 metastatic gastrointestinal tract cancer patients treated with ICIs and observed that the levels of serum monocyte chemoattractant protein 1 (MCP-1) in all patients with HPD were significantly lower than those in patients without HPD. Because MCP-1 has the ability to attract monocyte aggregation, we reasonably speculate that the lack of monocytes may promote the occurrence of HPD.

## 5. Macrophages

Macrophages are plastic and can transform the immune environment into a protumor or antitumor environment by releasing inflammatory or inhibitory cytokines and chemokines [10]. In addition, macrophages express both PD-1 and PD-L1 in the TME and are therefore likely to be constrained by anti-PD-1/PD-L1 antibody therapy. The levels of PD-1 and PD-L1 expression in macrophages have been shown to be critical for antitumor responses in preclinical models [84].

### 5.1. M2 Macrophages

As shown in Figure 2, Lo Russo et al. [10] provided evidence for the reprogramming of macrophages from the M1 to M2 phenotype, leading to HPD, by studying 187 NSCLC patients who received anti-PD-1/PD-L1 treatment. The researchers observed that all HPD patients had significant numbers of CD163^+^CD33^+^PD-L1^+^ M2 macrophages infiltrating their tumors. This finding suggested that macrophages were involved in the progression of HPD. To prove the role of tumor-associated macrophages (TAMs) in inducing HPD, researchers also conducted in vivo experiments using multiple immunodeficient NSCLC mouse models. They treated the mice with anti-mouse PD-1 antibodies and observed increased tumor growth compared to the controls, and the macrophage abundance in the TME was similar to that observed in HPD patients. As in tumor patients, specific macrophages present in the TME may affect the development of HPD through mechanisms that do not involve the direct blockade of PD-1 in immune cells prior to the initiation of anti-PD-1 antibody therapy. In addition, other researchers have also observed that, in the treatment of NSCLC, cervical cancer, breast cancer, and colorectal cancer, anti-PD-L1 treatment may trigger the accumulation of immunosuppressive M2 macrophages characterized by a CD163^+^PD-L1^+^ phenotype at the tumor site and the accumulation of M2 macrophages may worsen the prognoses of patients treated with ICIs [85,86,87,88].

In addition, higher serum levels of ANGPT2 before ICI treatment were associated with higher levels of immunosuppressive M2 macrophages in patients treated with ICIs [89,90]. An increase in serum ANGPT2 was also observed in patients with advanced melanoma treated with anti-PD-1 antibodies [90]. Such a high level of ANGPT2 is related to an increase in the number of M2 macrophages. Therefore, ANGPT2 has been suggested as a predictor and prognostic marker in advanced melanoma patients treated with ICIs [89,90]. In addition, Wang et al. [91] demonstrated that tumor-derived exosomes induce IL-10 production in PD-1^+^ macrophages and block CD8^+^ T-cell function. Both IL-10 and ANGPT2 are contained in transcriptional signatures associated with resistance against anti-PD-1 antibodies in melanoma patients, indicating that they rapidly act on the immune regulatory mechanism and can promote immune escape and, ultimately, tumor growth [92].

### 5.2. FcγRIIb

Antibodies consist of the F(ab’)_2_ fragment, which binds antigens and the fragment crystallizable (Fc) region, which is recognized by immunoglobulin Fc receptors (FcRs) expressed by immune cells [93,94]. The interaction between the Fc domain of an antibody and its associated FcR triggers a series of immune events [95]. Most ICIs are developed by selecting antibody isotypes with low binding affinity for FcRs in order to prevent killing T cells expressing immune checkpoints [96,97,98]. In fact, both nivolumab and pembrolizumab anti-PD-1 antibodies are the IgG4 type, which is an isotype considered immunologically inert. However, they still retain the ability to bind FcγRI and FcγRIIb to activate and inhibit FcRs, respectively [96,97,98].

TAMs are capable of capturing anti-PD-1 antibodies from T-cell membranes [99]. Lo Russo et al. [10] demonstrated that nivolumab-based Fab constructs lacking Fc components did not produce HPD-like results in experimental models. The significant tumor growth observed with the full antibody was completely abolished by the use of the anti-PD-1 antibody F(ab’)_2_ fragment and macrophage exhaustion. In animal studies, nivolumab-associated HPD showed the infiltration of M2-like macrophages, which was suspected to be caused by the binding of Fc (nivolumab)-FCγRs. The induction of HPD by TAM reprogramming depends on the interaction of the antibody Fc with the macrophage FcR rather than the anti-PD-1 F(ab)_2_ fragment in the environment of tumors treated with ICIs. Since anti-PD-1 antibodies bind to the suppressive receptor FcγRIIb, researchers have hypothesized that TAMs expressing a specific subset of the suppressive receptor FcγRIIb interact with the Fc domain of anti-PD-1 antibodies to undergo functional reprogramming, thereby acquiring enhanced tumor-promoting properties and ultimately inducing HPD. Thus, anti-PD-1 antibody treatments with specific Fc sequences have the potential to promote the reprogramming of macrophages into an aggressive phenotype, leading to HPD. As shown in Figure 3, Knorr et al. [100] described how anti-PD-1 antibodies established a connection between PD-1^+^ macrophages and FcγRIIb^+^ immune cells. This interaction is speculated to induce the aggregation of several PD-1 proteins on the macrophage membrane. Close contact between different PD-1 proteins may trigger PD-1-mediated signal transduction, leading to the polarization of macrophages toward the protumor phenotype. These findings suggest that HPD can be addressed by modifying current antibody therapies aimed at disabling FcR interactions.

Although FcγRIIb may cause HPD via at least two different mechanisms, the involvement of other FcRs should not be ruled out. For example, Swisher et al. [101] reported that IgG4 could limit the response of human single-cell-derived macrophages to IFN-γ via FcγRI interactions. Activation of IgG4- mediated FcγRI shifts macrophages toward an M2-like phenotype.

## 6. DCs

DCs are the most effective antigen-presenting cells (APCs) that induce an effective adaptive immune response and are therefore considered to be a key factor in determining the outcomes of ICI treatment [102]. Among them, tumor infiltrating DCs (TIDCs) are a heterogeneous group of bone marrow immune cells with different mature states and functions. Some subpopulations have immunostimulatory properties and can promote antitumor immunity, while other subgroups are immunosuppressive and can facilitate tumor immune survival and tumor progression [103,104].

### 6.1. PD-1

PD-1 is also expressed in DCs, and the infiltration of PD-1^+^ TIDCs in the TME has been observed [105]. PD-1 is thought to maintain DCs in an immature phenotype and inhibit cytokine production, costimulatory molecule expression, and the antigen presentation capacity [105,106]. PD-1 expression in DCs is regulated by different stimulus signals, such as IL-10 [79]. Additionally, compensatory immune responses induced by ICB can induce the production of immunosuppressive cytokines. The blockade of PD-1^+^ DCs promotes increased IL-10 release from these DCs, which further enhances the regulation of PD-L1 expression in DCs, thus forming a vicious cycle that ultimately leads to enhanced immunosuppression [79]. Zhao et al. [107] demonstrated that PD-1 and PD-L1 interact simultaneously on the same cell membrane in TIDCs. In this way, PD-1 blocks PD-L1 on the same cell, thus neutralizing its ability to bind T-cell PD-1. After the administration of anti-PD-1 antibodies, PD-L1 can again be used to induce the immunosuppressive activity of T cells.

### 6.2. PD-L2

PD-L2 is mainly expressed in macrophages and DCs [108]. The affinity between PD-L2 and PD-1 is higher than that between PD-L1 and PD-1 [109]. The differences in clinical efficacy and toxicity between anti-PD-1 and anti-PD-L1 antibodies observed in clinical studies is likely related to the fact that anti-PD-1 antibodies can block the binding of PD-1 and PD-L2 in addition to blocking the binding of PD-1 and PD-L1. Although these ligands compete for PD-1 receptors, they are likely to initiate different responses. The role of PD-L2 has not been as thoroughly explored as that of PD-L1, but ICB may affect the interaction of PD-1 with its two ligands in different ways [110].

While PD-L1 can interchangeably bind to PD-1 or CD80 (a membrane antigen necessary for T cell activation) in T cells, PD-L2 does not bind to CD80 and induces an inhibitory TME around T-cell receptors [111]. The blockade of PD-1 may lead to the CD80/PD-L1-mediated inhibition of T cells, while PD-L1 blockade may lead to the PD-1/PD-L2-mediated inhibition of T cells [112]. The blockade of the PD-1/PD-L2 interaction leads to significant loss of inducible co-stimulator (ICOS) and CD3 expression in CD4^+^PD-1^+^ T cells, suggesting that PD-1/PD-L2 blockade disrupts the immune response in some situations [113]. Administration of soluble PD-1-Ig fusion protein inhibited bone marrow-derived DC activation and increased IL-10 secretion. The preneutralization of PD-1 with anti-PD-1 antibodies prevented this effect, suggesting that the effect is PD-1-specific and mediated by PD-L1 or PD-L2 [114].

## 7. Neutrophils

In a clinical research of advanced gastric cancer (AGC), Sasaki et al. [14] treated 64 patients with nivolumab and observed that C-reactive protein (CRP) and absolute neutrophil count (ANC) levels were significantly increased only in patients with HPD. They speculated that the higher CRP and ANC levels in patients with HPD might be related to the strong release of neutrophils in bone marrow and the accumulation of myeloid-derived suppressor cells (MDSCs) in tumors. In addition, abnormal inflammation observed in patients with melanoma and prostate cancer treated with anti-PD-1/PD-L1 antibodies caused by the increased Th1/Th17-dependent secretion of inflammatory cytokines, such as IL-6, IL-17, and IFN-γ, has been shown to be associated with neutrophils [115]. In addition, neutrophil exhaustion and IL-6 blockade have been shown to lead to a more effective antitumor immune response in mice [116,117,118]. Lung adenocarcinoma transgenic mice overexpressing IL-17α have stronger intrinsic resistance to anti-PD-1 antibodies, which have been demonstrated, whereas anti-IL-6 antibodies or neutrophil exhaustion lead to an effective antitumor immune response [119]. It is important to note that neutrophils generally dominate the TME in NSCLC, and their presence is closely associated with reduced infiltration of CD8^+^ T cells into the tumor [120]. The primary role of IL-6 and IL-17 is to promote neutrophil-mediated inflammation [121,122]. The current evidence suggests that PD-1/PD-L1 inhibition may regulate the IL-6/IL-17-neutrophil axis, resulting in abnormal inflammation, tumor immune escape, and accelerated tumor growth. Therefore, the aggregation of neutrophils in tumors after ICI treatment may lead to the generation of an immunosuppressive environment and thus promote the occurrence of HPD.

Since neutrophil infiltration may be associated with worse treatment outcomes, assessing neutrophil infiltration may be an important parameter for predicting immunotherapy outcomes [123]. One of the simplest methods is to analyze the neutrophil to lymphocyte ratio (NLR) in peripheral blood. The NLR is commonly used to assess general immune responses and systemic inflammatory states [124,125] but has also shown excellent clinical utility in ICI therapy. Mezquita et al. [126] analyzed the derived NLRs (dNLRs) in patients with NSCLC treated with chemotherapy or ICIs. A high number of circulating neutrophils was thought to be independently associated with adverse outcomes of ICI therapy but not of chemotherapy. Similar results were reported in metastatic urothelial carcinoma, and the NLR was a significant predictor of HPD [127]. Ferrara et al. [128] observed a significantly higher percentage of circulating immature low-density neutrophils in HPD patients than in non-HPD patients before ICI treatment. In addition, Gr1^high^Ly6C^low^IL-17^+^ neutrophil subsets were observed in mice treated with anti-PD-1 antibodies and exhibiting HPD-like tumor growth. These data suggested that neutrophils might be another indispensable factor that could promote HPD because an increased number of tumor-infiltrating neutrophils might affect the function and number of T cells, leading to a severely impaired ICI therapeutic effect.

## 8. MDSCs

MDSCs are a group of heterogeneous cells derived from bone marrow. They are precursors of DCs, macrophages, and granulocytes and have the ability to significantly inhibit the immune cell response [129]. PD-1 blockade negatively interacts with the innate immune system. Furthermore, MDSCs have high PD-L1 expression in the TME and weaken the efficacy of ICIs by competing with anti-PD-1 antibodies to bind T cells and secrete immunosuppressive molecules [130]. Lo Russo et al. [10] reported that MDSCs contribute to HPD, especially in the presence of M2 macrophages. In addition, IFN-γ was significantly increased after PD-1/PD-L1 blockade [131]. IFN-γ has been proven to stimulate the development of MDSCs [90] and induce the expression of indoleamine 2,3-dioxygenase, which induces the Treg cell regulation of immunosuppression in vivo [47,67]. This finding is consistent with the presence of a large number of MDSCs in the peripheral blood of HPD patients [132].

## 9. Natural Killer (NK) Cells

NK cells contribute to the immune attacks against tumors, but their cytotoxicity is affected by the expression level of PD-1. Anti-PD-1 antibodies not only induce the generation of powerful T cells but also restore antitumor responses in NK cells [117,133]. Solaymani-Mohammadi et al. [134] observed that a lack of PD-1 reduces the expression levels of effector molecules in NK cells, such as granzyme B and perforin, limiting the antitumor function of NK cells. Thus, complete PD-1 signaling is necessary to maintain the full function of NK cells. When PD-1 is blocked, researchers have observed the reduced production of lactic acid molecules, such as perforin and granzyme in NK cells. Therefore, NK cells may also be potential contributors to HPD.

## 10. Innate Lymphoid Cells (ILCs)

ILCs are a group of innate immune cells that contribute to lymph node development, tissue damage repair, antimicrobial infection, and other processes. Their dysfunction is closely related to allergic diseases, chronic infections, metabolic diseases, tumors, and other diseases [135]. ILCs are divided into several subgroups according to their characteristic surface markers, transcription factors, secreted cytokines, and immunoregulatory effects. ILC1s and ILC3s can mediate the immune response against viral, bacterial, and parasitic infections, while ILC2s are essential for maintaining tissue homeostasis [136]. The roles of these cells have been popular research topics in the field of immunology in recent years. ILCs are involved in the regulation of the innate and adaptive immune responses by secreting cytokines upon activation by different stimuli [137]. Interestingly, the immunomodulatory function of ILCs depends on secreted cytokines and the TME; thus, they play a dual role in antitumor immunity [138].

ILC3s can promote tumor progression by secreting IL-17, IL-22, and granulocyte-macrophage colony-stimulating factor (GM-CSF) [139]. Kirchberger et al. [140] reported that IL-22 produced by ILC3s reduced gastrointestinal cancer growth in colon cancer-model mice. Irshad et al. [141] provided evidence for an association between the presence of ILC3s in the TME and an increased risk of lymph node metastasis in breast cancer-model mice. Carrega et al. [142] proved that ILC3s express natural cytotoxic receptors (NCRs), which are generally considered to be expressed only in NK cells. These NCR^+^ ILC3s contribute to the formation of tertiary lymphoid structures (TLSs) involved with advanced tumors in patients with NSCLC. Xiong et al. [143] analyzed two patients with HPD mediated by anti-PD-1 antibodies and observed the enrichment of ILC3 marker genes. It has been reported that PD-1 is also expressed in ILC3s [144,145]. Therefore, anti-PD-1 antibodies may induce the reactivation of ILC3s and the subsequent release of cytokines, which may have protumor activity in some types of tumors [139,140]. In addition, Hepworth et al. [146] reported that ILC3s inhibited the antitumor responses of T cells by competing with T cells for IL-2, further enhancing the immunosuppression of the TME.

## 11. Conclusions and Future Perspectives

The advent of immunotherapy in oncology has led to unprecedented improvements in the outcomes of patients with multiple tumor types. However, evidence has indicated that immunotherapy might have a deleterious effect in some tumor patients, even promoting their disease progression. At present, many existing clinical, biological, and histopathological data that prove that a variety of immune cells modulate the effect of immunotherapy in complex ways. These findings enrich the theoretical basis of immunotherapy. Since the potential mechanisms of HPD remain uncertain, the exploration of the causative mechanisms behind HPD and the development of detection and prediction methods are urgently needed, especially in the increasingly complex clinical trials applying multiple ICIs together or an ICI in combination with other tumor treatment approaches. Further research is necessary to confirm the hypotheses that have been proposed so far and to explore other potential molecular and immunological bases of HPD to identify clear predictive factors and possibly prevent HPD. Therefore, more efforts are required in the future to inform clinical decision making for patients undergoing immunotherapy.

## Figures and Tables

**Figure 1 cells-11-01758-f001:**
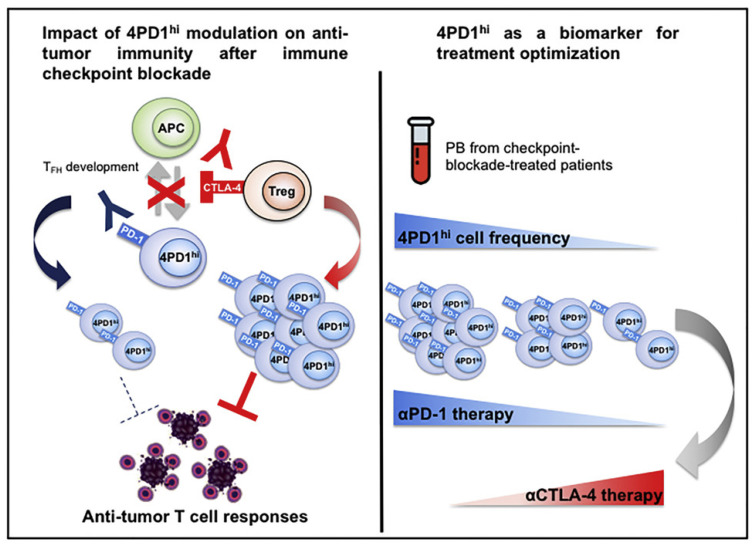
Zappasodi et al. illustrated the function of 4PD1^hi^ cells (PD-1^+^CD4^+^Foxp3^−^ T cells) and observed that these cells accumulate within the tumor as a function of the tumor burden. Reprinted with permission from Ref. [61]. 2018, Elsevier.

**Figure 2 cells-11-01758-f002:**
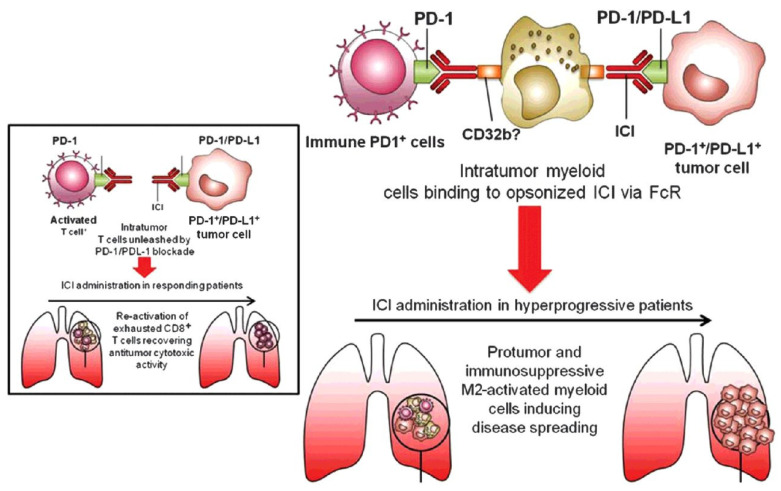
Hypothesized mechanism through which macrophages and ICIs are involved in HPD development. Reprinted with permission from Ref. [10]. 2019, American Association for Cancer Research.

**Figure 3 cells-11-01758-f003:**
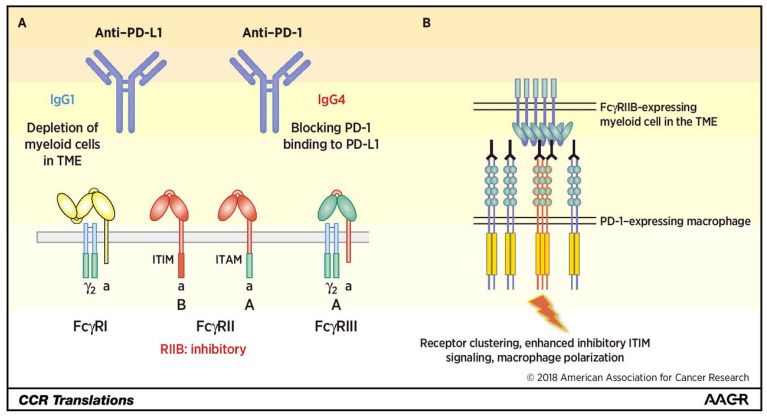
(**A**) Anti-PD-L1 antibodies (IgG1) and anti-PD-1 antibodies (IgG4) have different molecular mechanisms exerted through binding with activating or inhibitory FcRs. (**B**) FcyRIIB-enhanced clustering of PD-1 on macrophages results in polarization via enhanced signaling based on the immunoreceptor tyrosine-based inhibitory motif (ITIM) domain. Reprinted with permission from Ref. [100]. 2019, American Association for Cancer Research.

**Table 1 cells-11-01758-t001:** Representative case series studies of HPD in multiple cancer types.

Reference	Year	Country	Cancer Types	Incidence, *n* (%)	Clinical Relevance
Champiat et al. [6]	2017	France	Pan-cancer	12/131 (9%)	HPD was associated with a higher age (>65 years old) and a worse OS, but not associated with higher tumor burden or any specific tumor type.
Saâda-Bouzid et al. [7]	2017	France	R/M HNSCC	10/34 (29%)	HPD significantly correlated with a regional recurrence, but not with local or distant recurrence. HPD was associated with a shorter PFS, but not with OS.
Ferrara et al. [8]	2018	France	NSCLC	56/406 (13.8%)	HPD was associated with high metastatic burden and poor prognosis.
Kanazu et al. [9]	2018	Japan	NSCLC	5/87 (5.7%)	HPD was thought to be associated with poor quality of life and survival.
Lo Russo et al. [10]	2019	Italy	NSCLC	39/152 (25.7%)	Pretreatment tissue samples from all patients with HPD showed tumor infiltration by M2-like CD163^+^CD33^+^PD-L1^+^ clustered epithelioid macrophages.
Kim et al. [11]	2019	South Korea	R/M NSCLC	55/263 (20.9%)	A lower frequency of effector/memory subsets (CCR7^−^CD45RA^−^ T cells among the total CD8^+^ T cells) and a higher frequency of severely exhausted populations (TIGIT^+^ T cells among PD-1^+^CD8^+^ T cells) in peripheral blood were associated with HPD and inferior survival rate.
Aoki et al. [12]	2019	Japan	AGC	19/100 (19.0%)	HPD was observed more frequently after nivolumab compared with irinotecan, which was associated with a poor prognosis after nivolumab but not so clearly after irinotecan.
Kanjanapan et al. [13]	2019	Canada	Pan-cancer	12/182 (6.6%)	HPD was not associated with CSAEs, age, tumor type, or the type of immunotherapy but was more common in females.
Sasaki et al. [14]	2019	Japan	AGC	13/62 (21.0%)	Elevations in ANC and CRP levels upon nivolumab treatment might indicate HPD.
Lu et al. [15]	2019	China	Metastatic GTC	5/56 (8.9%)	Baseline serum levels of MCP-1, LIF, and CD-152 were associated with HPD.
Matos et al. [16]	2020	Spain	Pan-cancer	29/270 (10.7%)	The HPD progressor group had a significantly lower OS when compared with the non-HPD progressor group.
Castello et al. [17]	2020	Italy	NSCLC	14/46 (30.4%)	HPD status was associated with tumor burden. The derived neutrophil-to-lymphocyte ratio and platelet count were significantly associated with HPD status.
Hwang et al. [18]	2020	South Korea	RCC/UC	13/203 (6.4%)	HPD developed predominantly in patients with UC, and the incidence of HPD in patients with RCC was negligible. UC and creatinine above 1.2 mg/dL were independent predictive factors for HPD. A 30% increase in lymphocyte number following PD-1/PD-L1 inhibitor treatment was a negative predictor of HPD.
Vaidya et al. [19]	2020	USA	Advanced NSCLC	19/109 (17.4%)	Image-based radiomics markers extracted from baseline CTs might help identify patients at risk of HPD.
Zhang et al. [20]	2020	China	Advanced HCC	10/69 (14.5%)	Haemoglobin level, portal vein tumour thrombus, and Child-Pugh score were significantly associated with HPD. Patients with HPD had a significantly shorter OS than that of the patients with non-HPD.
Kim et al. [21]	2021	South Korea	Advanced HCC	24/189 (12.7%)	Patients with HPD had worse PFS and OS compared to patients with progressive disease without HPD. An elevated neutrophil-to-lymphocyte ratio (>4.125) was associated with HPD and an inferior survival rate.
Chen et al. [22]	2021	China	Pan-cancer	38/377 (10.1%)	Patients with HPD had lower OS than those without HPD. KRAS status was significantly associated with HPD in patients with colorectal cancer. The rapid increase of characteristic tumor markers within 1 month was associated with the occurrence of HPD.
Xiao et al. [23]	2021	China	PLC	13/129 (10.1%)	The PFS of HPD patients was significantly worse than that of non-HPD patients. Compared with the non-HPD patients, lung metastasis, and lymph node metastasis were independent risk factors of HPD.
Miyama et al. [24]	2021	Japan	UC	6/23 (26.1%)	Squamous differentiation may be a novel biomarker for predicting HPD in patients with UC who receive pembrolizumab.
Maesaka et al. [25]	2022	Japan	Unresectable HCC	9/88 (10.2%)	Patients with HPD had larger and more intrahepatic tumors, higher levels of α-fetoprotein and lactate dehydrogenase, and higher NLR at baseline than patients without HPD. NLR of ≥3 at baseline was identified as the only independent factor associated with HPD in multivariate analysis.

R/M, recurrent, and/or metastatic; HNSCC, head, and neck squamous cell carcinoma; NSCLC, non-small cell lung cancer; AGC, advanced gastric cancer; GTC, gastrointestinal tract cancer; RCC, renal cell carcinoma; UC, urothelial carcinoma; HCC. hepatocellular carcinoma; PLC, primary liver cancer; OS, overall survival; PFS, progression-free survival; CSAEs, clinically significant adverse events; ANC, absolute neutrophil count; CRP, C-reactive protein; MCP-1, monocyte chemoattractant protein 1; LIF, leukemia inhibitory factor; CD-152, cluster of differentiation 152; NLR, neutrophil-to-lymphocyte ratio.

## Data Availability

Not applicable.

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
