# Peer review of "Immune Cells in Hyperprogressive Disease under Immune Checkpoint-Based Immunotherapy"

_cells, 2022, doi:10.3390/cells11111758_

Round 1

Reviewer 1 Report

This is a short review about cellular resistance related to immune checkpoint block therapy. The manuscript is reasonably structured and well written. But minor revisions are necessary to improve it. Figure 4 is missing. Rapid progression of adult T-cell leukemia-lymphoma has been reported after anti-PD-1 therapy (N Engl J Med. 2018 May 17;378(20):1947-1948; Blood. 2019 Oct 24;134(17):1406-1414), which was not discussed. 

1. The authors copy all three figures from the original publications. 2. Proofreading is necessary, such as "one ICIs" in line 29 and "HDP" in line 381. 3. It would be ideal to include a table demonstrating the incidence of Hyperprogressive Disease (HPD) related to immune checkpoint blockade treatment in human cancers based on recent clinical trials.

Reviewer 2 Report

The topic covered by this review is interesting, but there are similar recently published articles in this field that seem more detailed and appropriate.

The authors reported a list of studies and results, without a real discussion of the data.

The figures are not original, all three figures are taken from other publications. Do the authors have permission?

Figure 1 was taken from reference 38. The caption is a sentence directly copied from the abstract of reference 38.

Figure 2 was taken from reference 64

Figure 3 and its caption were taken from reference 81

Specific comments:

Line 29: "one ICIs" should be "one ICI"

Lines 56-59: the authors stated in lines 56-57 that the response to ICI is limited by infiltration of T cells and in lines 57-59 the opposite.

Lines 59-61: it is possible, but not "reasonable" to assume that CD8, having a positive role in the treatment of ICI, must logically also have the opposite effect.

Paragraph 2.1:

Authors should introduce and discuss the topic of exhausted T lymphocytes. For reference, see “Jiang et al. Front Immunol 2021 ".

Line 74 and in other part of the manuscript: Avoid drug brand names

Line 777: TIL is not specified

Paragraph 3:

Line 123: What does " they are not only crucial for promoting CD8(+) T cell help" mean? CD4 T cell help has a role in promoting CD8 functions, preventing CD8 depletion or inducing CD8 memory, etc ...

Paragraph 3.1

Regulatory T lymphocytes play a role not only in autoimmunity but in a wide range of diseases (allergies, chronic infections, parasitic infections, etc ...)

Paragraph 3.3

Introduce and deeply discuss the most recent literature on the subject. For example, "Miggelbrink et al Clinical Cancer Research 2021". Also of interest: "Balancaga et al Journal of Clinical Investigation 2020".

Lines 188-190: the cited reference does not seem to deal with the data reported in the text, as it relates to the depletion of T lymphocytes in viral infections

Paragraph 3.4

The role of IFN-gamma in promoting tumor progression is related, in relation to reference 45, to its inability to control papilloma infection and to promoting Th17 inflammation. Authors should go deeper in reporting data of the cited works to allow a complete understanding of the issue.

Reference 47, discussed in lines 197-200, does not deal with mycobacterium tuberculosis infection. It addresses the role of myeloid suppressor cells in tumor progression, where IFN-gamma plays an important role.

Paragraph 4 and followings: the data are poorly discussed and sometimes incomplete.

Paragraph 10: Are the authors sure that ILC2s are dysfunctional in allergic diseases?

Reviewer 3 Report

In this review, Wei and Zhang summarizes recent finding on the role of the immune system cells in hyperprogressive disease in response to immune checkpoint therapy in cancer. It is timely and deserves publication in Cells. Only a few minor considerations

1.- A table summarising the expression pattern of PD-1 in the different immune cell populations and the impact of its interaction with anti-PD-1 antibodies on their functionality could help.

2.- Instead of indicating the expression of a given marker in the form (+), (e.g. CD8(+)), it would be more appropriate to indicate it as a super-index (e.g. CD8+).

3.- Line 381: HPD instead of HDP
